# Identification of Proteins Specifically Assembled on a Stem-Loop Composed of a CAG Triplet Repeat

Robert P. Fuchs [1,2,*], Asako Isogawa [3], Joao A. Paulo [4] and Shingo Fujii [3,*]

1 Marseille Medical Genetics, UMR1251, 13005 Marseille, France
2 Department of Biological Chemistry and Molecular Pharmacology, Harvard Medical School, Boston, MA 02115, USA
3 Cancer Research Center of Marseille, CNRS, UMR7258, 13009 Marseille, France; asako.isogawa@inserm.fr
4 Department of Cell Biology, Harvard Medical School, Boston, MA 02115, USA; joao_paulo@hms.harvard.edu
* Correspondence: roberpipo@gmail.com (R.P.F.); shingo.fujii@inserm.fr (S.F.)

**Abstract:** Human genomic DNA contains a number of diverse repetitive sequence motifs, often identified as fragile sites leading to genetic instability. Among them, expansion events occurring at triplet repeats have been extensively studied due to their association with neurological disorders, including Huntington's disease (HD). In the case of HD, expanded CAG triplet repeats in the *HTT* gene are thought to cause the onset. The expansion of CAG triplet repeats is believed to be triggered by the emergence of stem-loops composed of CAG triplet repeats, while the underlying molecular mechanisms are largely unknown. Therefore, identifying proteins recruited on such stem loops would be useful to understand the molecular mechanisms leading to the genetic instability of CAG triplet repeats. We previously developed a plasmid DNA pull-down methodology that captures proteins specifically assembled on any sequence of interest using nuclear extracts. Analysis by Mass Spectrometry revealed that among the proteins specifically bound to a stem-loop composed of CAG triplet repeats, many turned out to belong to DNA repair pathways. We expect our data set to represent a useful entry point for the design of assays allowing the molecular mechanisms of genetic instability at CAG triplet repeats to be explored.

**Keywords:** plasmid DNA pull-down; CAG triplet repeat; MS analysis





## 1. Introduction

Maintaining the integrity of genomic DNA is an essential demand for a healthy life in any species. However, specific DNA sequence contexts (e.g., triplet repeats, G-rich sequences inducing G-quadruplexes) are intrinsically fragile and potentially harmful during their lifespan [1,2]. Among them, genetic instability of triplet repeats is extensively studied because of their causality in hereditary or sporadic neurological diseases (e.g., Huntington's disease (HD), fragile X syndrome (FXS), myotonic dystrophy type 1 (DM1), amyotrophic lateral sclerosis (ALS)) [3]. For a long time, the pathogenesis of HD has been known to be closely correlated to the expansion of CAG triplet repeats in *HTT*, the gene encoding huntingtin protein. The onset of HD is thought to be mediated by the expanded polyglutamine stretches (CAG encodes glutamine) present in the mutant forms of HD protein leading to protein aggregation and the so-called notion of polyglutamine disorders. Such a fundamental premise for the onset of HD has, however, recently been challenged since the expanded length of CAG repeat per se was found to be positively correlated to the onset of HD rather than the length of the polyglutamine track in the HD protein. Indeed, it was shown that interrupting expanded CAG repeats by CAA triplets, that also encode glutamine residue, delays the timing of disease onset [4]. This finding indicates that the onset of HD directly relies on the length of the DNA (or RNA) uninterrupted CAG repeats per se rather than on the length of the polyglutamine tract in the resulting protein. In fact, such a possibility is likely to be shared by other DNA repeat expansion

diseases for which the culprit loci are located outside coding regions (untranslated region (UTR), introns) [3]. In order to tackle diseases involving structured DNA motifs such as HD, it is important to discover the proteins that bind to DNA intermediates that arise during the expansion process per se. For instance, it was proposed that the DNA repair enzyme OGG1 (7,8-dihydro-8-oxoguanine-DNA glycosylase), when acting on oxidized DNA damage located within a triplet repeat, can trigger repeat expansion via the formation of a CAG stem-loop intermediate structure in mouse cell lines [5].

From a global physiological point of view, it is known that normal *HTT* alleles that contain ~20 CAG repeats are translated into a correctly-folded protein and are stable in cellular populations. The transition from a healthy situation to a severe disease state involves several intergenerational and somatic CAG repeat expansion steps [3,6]. The initial CAG repeat expansion step, from 20 to 30–35 CAG repeats, is considered to represent an intermediate premutation state that is physiologically silent. When the premutated CAG allele further expands, it will gradually reach a disease-associated state as soon as the number of repeats exceeds 40 and up to 120. Based on recent advances in genome-wide association study (GWAS), it is suggested that many genes associated with the DNA damage response (DDR) substantially contribute to the CAG repeat expansion events [6,7]. Namely, key proteins belonging to the mismatch repair pathway [8] as well as FAN1 (DNA repair nuclease) [9] appear to be implicated as crucial players during CAG repeat expansion events.

We have developed a novel plasmid DNA pull-down methodology to capture specific nucleoprotein complexes from nuclear extracts in a DNA sequence-specific manner (termed IDAP: Isolation of DNA Associated Proteins), which turned out to be highly efficient and versatile [10,11]. *Xenopus* egg extracts are well known to support various DNA transactions (e.g., Replication, Repair, DNA damage response) [12–14]. For example, by using such extracts, IDAP was applied to capture and identify proteins recruited on DNA lesions induced by N-methyl-N-nitrosourea (MNU), a mimic of the chemotherapeutic agent temozolomide (TMZ) [15,16]. Based on the MS data and subsequent biochemical validation, we proposed that the cytotoxic effect of TMZ in nondividing cells may be triggered by a DNA double-strand break (DSB) generated via accidental encounter of base excision repair (BER) and mismatch repair (MMR) processes at closely-spaced lesions [15,16].

As mentioned above, it is believed that the critical feature of triplet repeats relies on their capacity to form stem-loops, when a triplet repeat sequence is exposed as single-stranded DNA during transactions such as replication, transcription, and repair [17]. As the plasmid DNA pull-down approach is an efficient protein discovery tool, we applied this approach to identify proteins that specifically bind to a CAG stem-loop in *Xenopus* egg extracts. For this purpose, we constructed a plasmid DNA possessing a stem-loop composed of a CAG triplet repeat on one strand [11]. In the present paper, using this newly constructed plasmid, we implemented the plasmid DNA pull-down approach to capture and identify specific proteins assembled on the stem-loop. We report characteristic features of the identified proteins, which appear to not only help us understand the molecular events occurring at the stem-loop but also provide potentially novel clues for developing effective treatments for patients.

## 2. Materials and Methods

### 2.1. Chemicals and Materials

Chemicals were from VWR (Rosny-sous-Bois, France) or Thermo Fisher Scientific (Illkirch, France), and all enzymes were from New England Biolabs (Evry, France). Oligo DNA and TFO were from Eurogentec (Seraing, Belgium). A TFO probe, TFO-1, was used [10]. This TFO-1 is composed of a psoralen residue, a 22-mer locked nucleic acid (LNA)/DNA mixed oligonucleotide, a spacer arm composed of tandemly oriented hexaethylene glycol, and a terminal desthiobiotin residue (see Figure 1 for its schematic drawing). The sequence context of the 22-mer oligonucleotide is 5′-tTtTcTtTtCtCCtCtTCtCct (LNA

and DNA residues are shown in small and capital letters, respectively). Capital C represents 5-methyl dC residue.

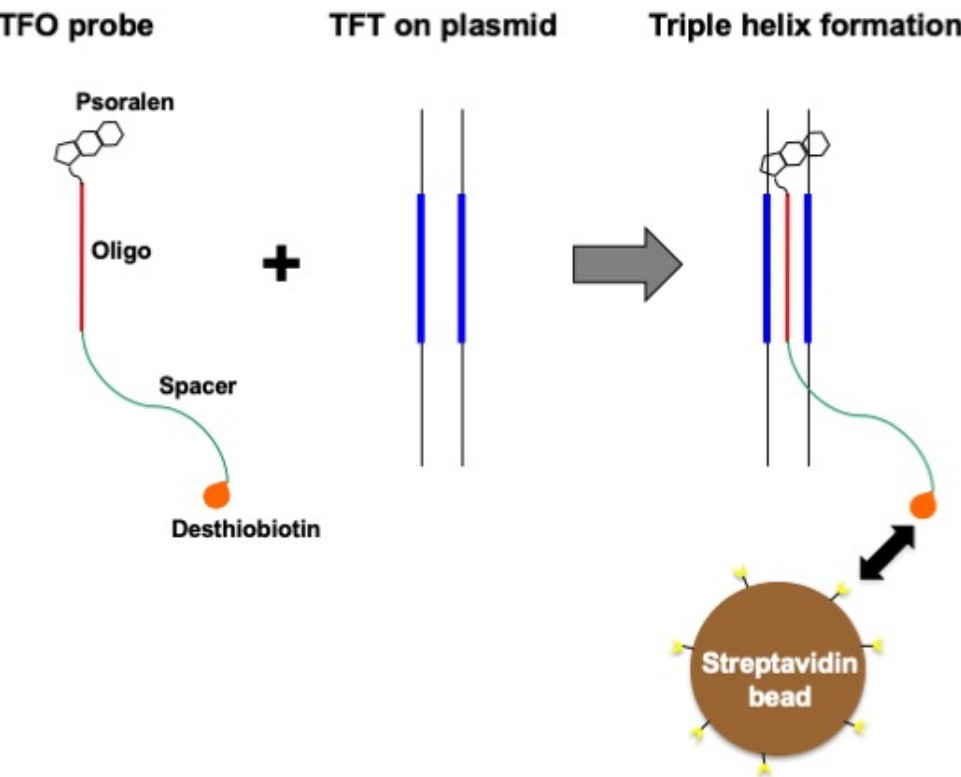

**Figure 1.** Strategy to capture a plasmid DNA via triple helix formation. TFO: Triple helix Forming Oligonucleotide. TFT: Triple helix Forming Tag.

*2.2. Plasmid DNA Substrates*

A CAG triplet repeat-containing plasmid (termed pAS203) is derived from a vector, pAS200.2, that contains a tag sequence, TFT-1, able to form a triple helix with a TFO-1 probe [11]. The sequence of TFT-1 is 5′-AAAAGAAAAGAGGAGAAGAGGA. Plasmid pAS203 contains a stretch of seven tandemly repeated CAG triplets surrounded by two nickase recognition sites (Figure 2). Three forms of pAS203 have been generated, a relaxed closed circular form (rcc), a gapped form (gap), and a closed stem-loop form (cSL), are termed pAS203(rcc), pAS203(gap), and pAS203(cSL), respectively (Figure 3). Detailed protocols to prepare all plasmid DNA substrates are described in [11].

*2.3. TFO-Conjugated Plasmid on Magnetic Bead*

Constructed plasmid substrates (250 ng of pAS203(rcc), pAS203(gap), or pAS203(cSL)) were conjugated with TFO-1 following triple helix formation as previously reported [10,11]. Each of the TFO-conjugated plasmids was mixed with 25 μL of Dynabeads M-280 Streptavidin (Invitrogen, Villebon-sur-Yvette, France) as in [10,11].

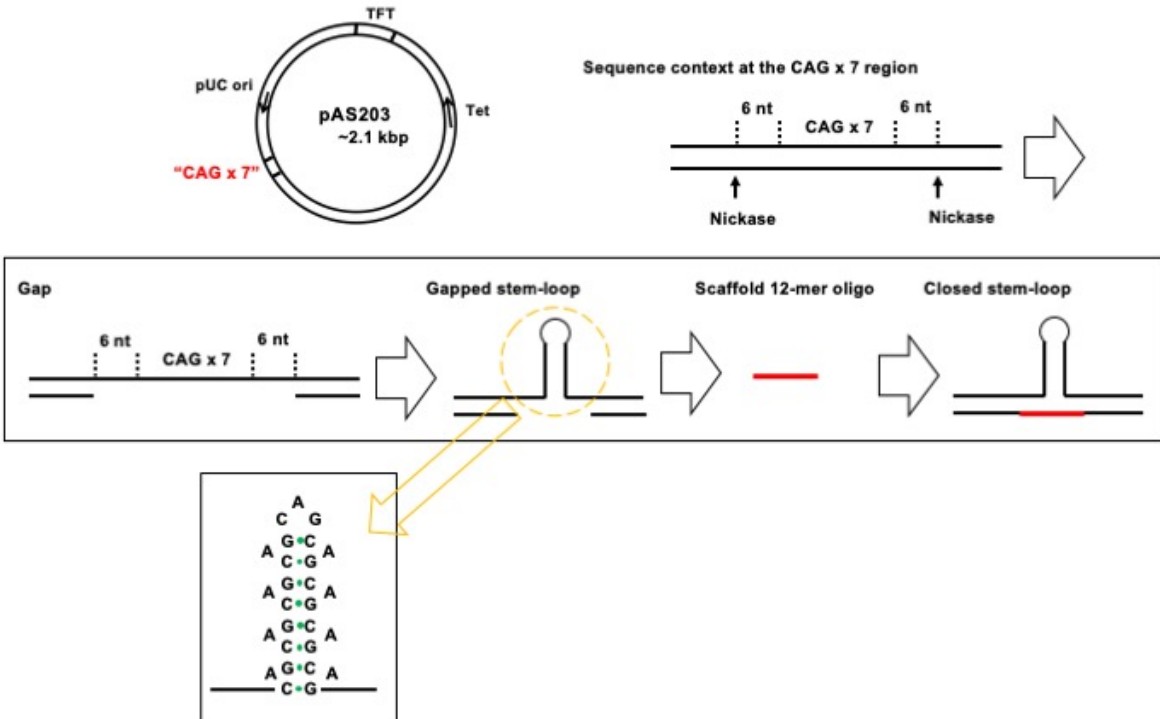

**Figure 2.** Construction of a plasmid harboring a CAG-containing gap and stem-loop. Two single-stranded nicks introduced in pAS203 within the same strand by two sequence-specific 'nickases' led to the excision of a short DNA fragment that can be released by heating (pAS203(gap)). Formation of a stem-loop composed of the CAG triplet repeat is achieved, following heat treatment, by ligation of a complementary oligonucleotide (12-mer) (pAS203(cSL)). pUC ori: high copy number pUC origin in *E. coli*. Tet: tetracycline resistance gene. CAG × 7: seven tandemly repeated CAG triplets.

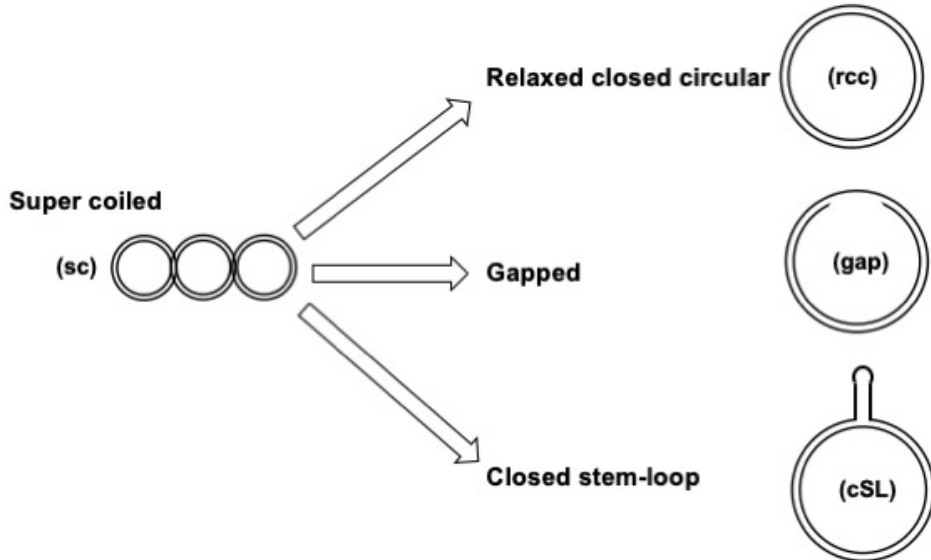

**Figure 3.** Constitution of DNA substrates. All three plasmids (rcc, gap, cSL) are derived from the same supercoiled (sc) plasmid. The rcc is constructed by self-ligation following the introduction of a nick by a nickase.

*2.4. Xenopus Egg Extracts*

We used nucleoplasmic extract (NPE) prepared from *Xenopus* eggs [18] as *Xenopus* egg extracts. The preparation of NPE is described in [19].

*2.5. Plasmid DNA Pull-Down and Protein Analysis*

Each of the TFO-conjugated plasmids (250 ng) immobilized on M-280 beads was incubated with 16 µL of NPE for 17 min at room temperature (RT) in the following buffer (10 mM HEPES-KOH (7.7), 50 mM KCl, 2.5 mM MgCl$_2$, 0.25 M sucrose, 0.1% NP40). Proteins bound on plasmid were crosslinked by adding 320 µL of a buffer containing 0.8% formaldehyde for 10 min at RT. Subsequent steps of bead wash and protein elution were the same procedure as previously reported [16]. The eluted proteins were analyzed by silver staining following polyacrylamide gel electrophoresis or by mass spectrometry (MS). For silver staining, an aliquot of each sample corresponding to 30 ng of plasmid was treated by heat to reverse the crosslinking condition. The samples were loaded and run on a 4–20% gradient gel (Bio-Rad, Marnes-la-Coquette, France) at 200 volts for 23 min and stained with a silver staining kit (Silver Stain Plus kit, Bio-Rad). MS and its data analysis were the same procedure as previously reported [16].

**3. Results**

Whereas precise molecular mechanisms of triplet repeat expansion are largely unknown, it has been presumed that stem-loop formation relying on intrinsic properties of triplet repeats would eventually trigger cascades leading to triplet repeat expansion (or contraction) in dividing or non-dividing cells [2,3]. In order to understand the molecular events occurring at stem-loops, the identification of proteins recruited on stem-loops will be instrumental in helping understand the mechanisms leading to genetic instability at triplet repeats.

*3.1. Assay Design to Capture and Identify Proteins Specifically Assembled on Structural DNA*

Our experimental goal is to identify proteins specifically recruited on a CAG triplet repeat stem-loop. As described in the introduction, we have developed a plasmid DNA pull-down methodology [10,11] to discover proteins that specifically associate with any DNA element of interest, such as CAG stem-loops. The core part of the methodology relies on the way the DNA element of interest is immobilized on magnetic beads. This is achieved by the use of a specific oligonucleotide (TFO probe) that forms a triple helix with a cognate double-stranded DNA (dsDNA) sequence (termed TFT: Triple helix Forming Tag) located in the same plasmid as the DNA element of interest (Figure 1). As the TFO probe carries a desthiobiotin moiety, the binary complex (i.e., dsDNA with the TFO probe) can be conjugated to a streptavidin-coated magnetic bead. As DNA substrates for the pull-down experiments, we constructed a plasmid pAS203 containing a stretch of seven tandemly repeated CAG triplet sequences, and flanked by two nickase recognition sites (Figure 2). By utilizing the nickases (Figure 2), three derivatives of pAS203 were constructed (Figure 3): a relaxed closed circular form, pAS203(rcc), a plasmid that contains the CAG repeat in its classical double-stranded form; a gapped form, pAS203(gap), a plasmid carrying a 33-nt long gap that encompasses the (CAG)$_7$ sequence; and a closed stem-loop form, pAS203(cSL) bearing a (CAG)$_7$ stem-loop [7]. These constructs were immobilized on beads and employed in the context of the plasmid DNA pull-down approach.

*3.2. Protein Capture in the Context of the Plasmid DNA Pull-Down Approach*

*Xenopus* egg extracts turned out to be particularly instrumental in the context of our plasmid pull-down methodology. Indeed, within the context of methylated DNA damage, we captured and identified, from frog extracts, specific proteins that specifically interact with alkylating lesions. These experiments led us to propose a novel mode of action of the drug temozolomide used in the treatment of glioblastomas [16]. Consequently, we decided to utilize *Xenopus* egg extracts to discover the proteins that specifically interact

with CAG-containing DNA intermediates related to HD. Each of the three constructed plasmids (i.e., pAS203(rcc), pAS203(gap), pAS203(cSL)) was individually incubated with a TFO-probe to form a triple helix with the cognate TFT sequence in the plasmid and then mixed with magnetic beads (Figure 4A). Subsequently, the TFO-conjugated plasmid beads were incubated with *Xenopus* egg extracts, followed by formaldehyde crosslink treatment. Subsequently, the beads were washed and eluted from the plasmid after cross-link reversion by heat. Eluted proteins are analyzed by electrophoresis on a silver-stained PAGE gel. Irrespective of the DNA substrates, the pattern of eluted proteins in lanes rcc, gap, and cSL (Figure 4B) revealed numerous proteins with a similar pattern of bands, a similar total amount, and a broadly distributed size range.

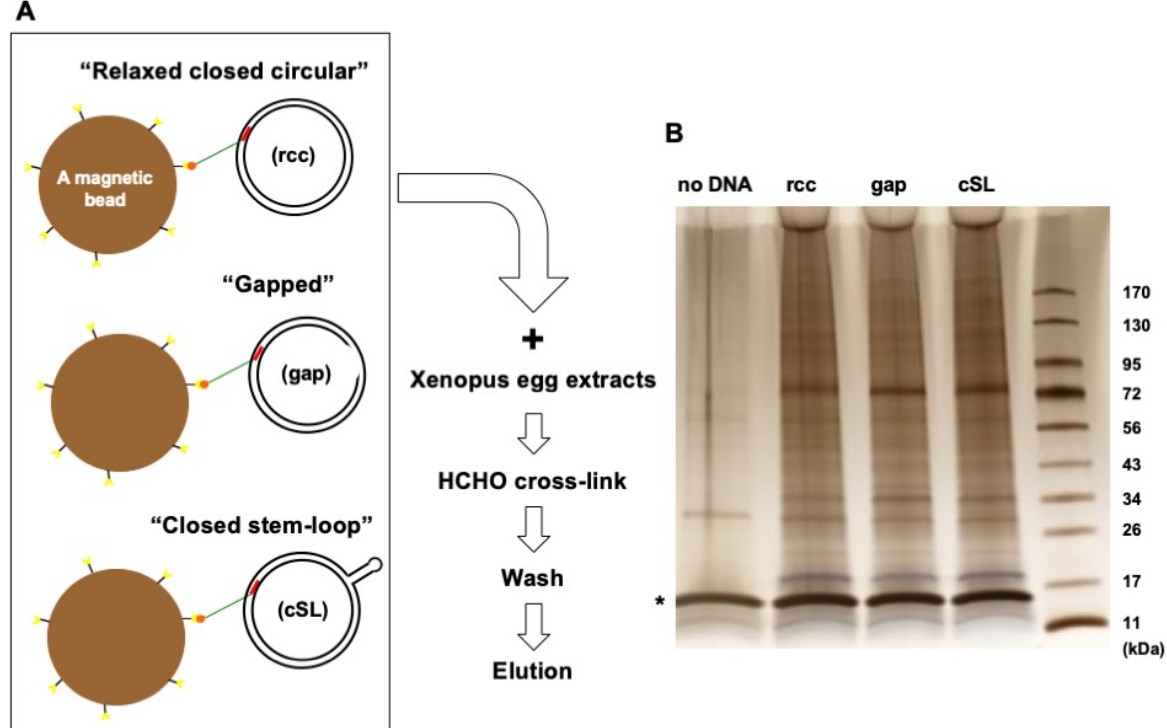

**Figure 4.** Protein capture by means of the plasmid DNA pull-down approach. (**A**) Schematic representation of the plasmid DNA pull-down approach. HCHO: formaldehyde. (**B**) Eluted proteins from 30 ng of each plasmid in the context of the plasmid DNA pull-down approach are loaded on a polyacrylamide gel following a reverse crosslink reaction. After electrophoresis, the gel is visualized by silver staining. The "no DNA" control lane is prepared in the same way but without the addition of plasmid DNA; *: streptavidin.

Importantly, in contrast to all other lanes, the "no DNA" control lane (Figure 4B) appears to be essentially free of proteins, reflecting low non-specific background protein carry-over. The high specificity of the plasmid DNA pull-down procedure assesses the high efficiency of the bead-washing procedure and guarantees optimal capture of specific DNA-binding proteins and their subsequent identification by MS analysis.

### 3.3. Identification of Proteins Specifically Recruited on the Stem Loop

Proteins bound to all three constructs (pAS203(rcc), pAS203(gap), pAS203(cSL)) were eluted from plasmid, 250 ng DNA equivalent each, and identified by MS analysis. As expected from the patterns observed in the silver-stained gel (Figure 4B), numerous proteins were identified by MS analysis (Table S1). In order to discover proteins that specifically bind to the CAG stem-loop structure, we compared the proteins eluted from the plasmid containing the stem-loop (pAS203(cSL) to the proteins eluted from the plasmid containing

the CAG sequence in its classical double-stranded form (pAS203(rcc)). A comparison was achieved by plotting the MS data in the form of a "Volcano" plot. Briefly, for each protein, the average spectral count value found in the stem-loop containing plasmid was divided by the average spectral count value in the corresponding double-stranded plasmid pAS203(rcc). The resulting ratio is plotted as its $\log_2$ value along the *x*-axis. On the other hand, the statistical significance of the data is roughly estimated by the *p*-value as determined in the Student's *t*-test and plotted as $-\log_{10}$ (*p*-value) along the *y*-axis. The IDAP approach represents a proteomic screening tool that associates our plasmid pull-down methodology with label-free MS analysis to obtain protein hits. These hits are conveniently identified as located on the right and left sides of the *y*-axis for enrichment and impoverishment, respectively (Figure 5). As for any screening test, the biological significance of the protein hits needs to be validated by additional experiments (biochemistry, genetics, etc.).

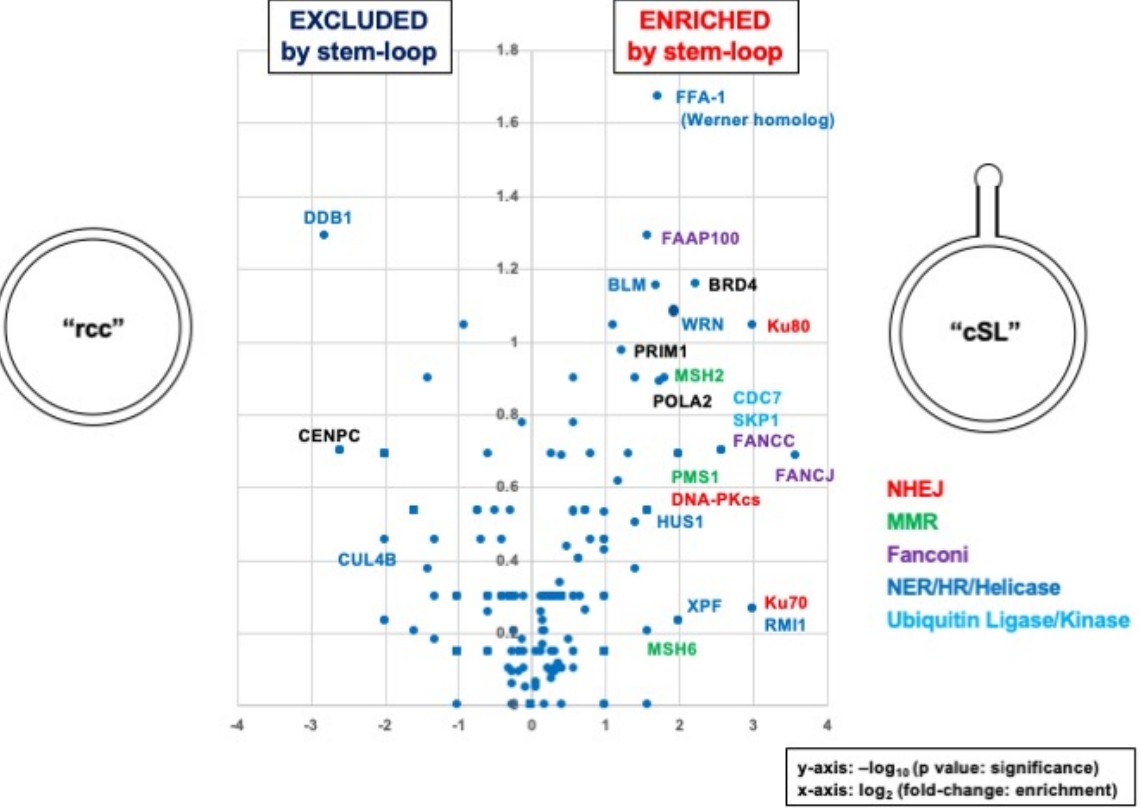

**Figure 5.** Identification of proteins assembled on the stem-loop. Proteins preferentially assembled on the stem-loop structure compared to fully double-stranded DNA are conveniently visualized via a volcano plot representation. Briefly, for each protein, the ratio between the average spectral count values on the stem-loop containing plasmid (pAS203(cSL)) and on the fully double-stranded plasmid (pAS203(rcc)) is plotted as its $\log_2$ value along the *x*-axis. Along the *y*-axis, we plot $-\log_{10}$ *p*-value as determined by Student's *t*-test. Proteins known to be associated with common repair pathways are highlighted in color fonts. All proteins identified in the MS analysis are listed in Table S1. *x*-axis: $\log_2$ (fold-change: enrichment); *y*-axis: $-\log_{10}$ (*p*-value: significance). NHEJ: non-homologous end joining. MMR: mismatch repair. NER: nucleotide excision repair. HR: homologous recombination.

Interestingly, among the enriched proteins, many appear to belong to well-characterized DNA repair pathways, such as non-homologous end joining (NHEJ), mismatch repair (MMR), and nucleotide excision repair (NER). In contrast, few proteins appear to be excluded by the stem-loop (left-side corner). Similarly, using the "Volcano" plot representation, we analyzed the proteins differentially bound to the gap-plasmid in comparison to

fully double-stranded DNA (Figure S1 for pAS203(rcc) vs. pAS203(gap)) or to the stem-loop construct (Figure S2 for pAS203(gap) vs. pAS203(cSL)).

## 4. Discussion

The onset of Huntington's disease (HD) is closely linked to the expansion of CAG triplet repeats in the *HTT* gene [3]. The importance of the length of the CAG triplet repeat to HD onset was demonstrated [4]. Ways to shorten expanded CAG triplet repeats would be instrumental in the context of effective therapies toward HD. Along these lines, it was found that a chemical compound, naphthyridine-azaquinolone (NA), specifically binds to stem loops composed of CAG triplet repeats but not to canonical dsDNA composed of CAG·CTG repeats. The structure-specific binding DNA compound NA was demonstrated to lead to the contraction of expanded CAG triplet repeats in HD mice models irrespective of the growing phases (nondividing or dividing) [20]. Of interest, such contraction events in vivo depend on the presence of functional MutSβ (a component of MMR) that recognizes small loops in dsDNA [21]. It was also reported that the introduction of TALEN (a transcription activator-like effector) leads to the contraction of expanded triplet repeats in model yeast strains [22]. For this contraction event, DNA double-strand break repair (DSBR)-related proteins were shown to be indispensable. Therefore, one potential therapeutic strategy toward alleviating HD is to harness intrinsic DNA repair pathways by artificially stimulating their activities.

The main goal of the present paper is to reveal clues that may prove to be instrumental in the design of novel and effective therapeutic strategies for HD. By utilizing the plasmid DNA pull-down approach, MS analysis revealed many interesting proteins specifically recruited by the presence of the stem-loop (Figure 5, Table S1). For example, the following DNA repair-related proteins were identified: NHEJ-related Ku70/80 and DNA-PKcs [23]; MMR-related MSH2, MSH6, and PMS1 [21]; Fanconi-related FAAP100, FNACC, and FANCJ [24]; NER/HR/Helicase-related WRN, BLM, XPF, HUS1, and RMI1 [25–27]. In human cell extracts, factors such as DNA-PKcs and Ku heterodimer, along with other NHEJ proteins, were previously identified by proteomics as binding to hairpins in single-stranded DNA [28,29].

Some "cSL" specific factors are commonly found in the "gap" containing plasmid (e.g., FANCC, MSH6, BLM, WRN); these proteins may be recruited on the stem-loop structure that can transiently form in the gapped plasmid (Figure S1, Table S1). When comparing "gap" with "cSL" (Figure S2, Table S1), some factors are enriched in "cSL" (e.g., Ku80, BRD4). It may indicate that recruitment of these factors specifically needs a stable stem-loop. In any case, our approach represents a proteomic screen for DNA-binding proteins that require additional biochemical and/or genetic experiments to validate the identified hits in terms of mechanisms and functions.

A limitation of our approach relates to the fact that we cannot ascertain if all specifically identified proteins are simultaneously recruited on a single plasmid DNA molecule. Nevertheless, from the present data set, we can derive molecular events that may occur at the stem-loop as a working model. As outlined in the Introduction, MMR proteins [8] and FAN1 [9] may be key players for CAG repeat expansion events. For instance, in the case of MMR-related proteins, several models were proposed in which MMR activities contribute to the expansion (or contraction) of triplet repeats [3,8,30]. In these models, MutSβ (MSH2/MSH3) and MutLα (MLH1/PMS2) access stem loops in a manner similar to the way they recognize small DNA loops in regular double-stranded DNA. On the other hand, we detected MSH2, MSH6, and PMS1 as MMR-related proteins but not MSH3 and PMS2. Namely, it may indicate that MutSβ and MutLα do not participate in the early phase of stem-loop recognition or processing. PMS1 is a component of MutLβ (MLH1/PMS1) the role of which is largely unknown in MMR [21]. As a working model, we can speculate that either MutLβ, PMS1 per se or a PMS1-containing complex associating with or without MutSα (MSH2/MSH6) may be adapted to recognize structural DNA such as the ones found in stem loops. Intriguingly, MSH6 and PMS1 are enriched in "gap" (Figure S1, Table S1). It

implies that these protein factors are recruited on unstable stem loops that will be formed in an early stage during events of genetic instability at CAG triplet repeats. Of interest, it should be noted that another putative key player FAN1 [9] apparently exhibits similar protein enrichment profiles with PMS1 in "gap" compared with "rcc" (Figure S1, Table S1) or with "cSL" (Figure S2, Table S1). On the other hand, differently from PMS1, FAN1 is not enriched in "cSL" compared with "rcc" (Figure 5, Table S1). The observation that FAN1 is preferentially recruited on DNA substrates with single-stranded/double-stranded junctions as in the "gap" substrate but not in closed substrates like "rcc" or "cSL", appears to be in good agreement with the proposed roles of FAN1 in DNA end processing during the CAG repeat expansion events [9]. Identification of unexpected (or unknown) proteins, thus, stimulates ideas for designing assays to explore new mechanisms.

While the present paper underscores the power of the IDAP approach for the identification of proteins that assemble on a short CAG hairpin, additional experiments will be necessary to obtain data sets that will be physiologically more relevant to the field of Huntington's disease. Future investigations will include the use of human cell extracts instead of frog extracts. We also intend to investigate the interactome when the CAG repeat is located within its actual *HTT* gene sequence context. Longer CAG expansion intermediates (up to the disease-causing allele of 37 CAG repeat), as well as the effect of interspersed CAA triplets, will also be investigated.

## 5. Patents

R.P.F. and S.F. hold a patent, 100007204, covering the conceptualization and methodology described in this manuscript.

**Supplementary Materials:** The following supporting information can be downloaded at: https://www.mdpi.com/article/10.3390/dna3020009/s1, Table S1: Resource data of MS analysis, containing three tabs; Figure S1: Volcano plot, rcc vs. gap; Figure S2: Volcano plot, gap vs. cSL.

**Author Contributions:** Conceptualization, R.P.F. and S.F.; methodology, R.P.F., A.I., J.A.P. and S.F.; formal analysis, R.P.F. and S.F.; data curation, R.P.F., A.I., J.A.P. and S.F.; writing—original draft preparation, S.F.; writing—review and editing, R.P.F. and S.F. All authors have read and agreed to the published version of the manuscript.

**Funding:** This research was funded by the National Institute of Health, grant number R01 GM132129 to J.A.P.

**Institutional Review Board Statement:** Not applicable.

**Informed Consent Statement:** Not applicable.

**Data Availability Statement:** All data were provided in the manuscript and Supplementary Materials.

**Acknowledgments:** We thank Johannes Walter (Harvard Medical School, USA) for providing space, support, and materials.

**Conflicts of Interest:** R.P.F. and S.F. are cofounders and consultants for bioHalosis, which provides a custom service of specific nucleoprotein capture used herein. The company had no role in the design of the study; in the collection, analyses, or interpretation of data; in the writing of the manuscript, or in the decision to publish the results. All other authors declare no conflict of interest.

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
