# Peer review of "Identification of Proteins Specifically Assembled on a Stem-Loop Composed of a CAG Triplet Repeat"

_2673-8856, doi:10.3390/dna3020009_

Round 1
Reviewer 1 Report
I appreciate the manuscript's interesting, novel, and elegant study design.
Although I do not have any reservations about the result of the study, there are two concerns that I must share about the manuscript.
The first concern is the lack of background information about the recent development of CAG repeat expansion research in Huntington's disease field. I encourage the authors to review articles in Volume 10, Issue 1 of the Journal of Huntington's Disease and incorporate them in the introduction as well as in the conclusion.
The second concern is the data itself. Figure 5 states that highlighted proteins are significantly enriched. I understand that the authors did not say it was significantly enriched in the body, but the figure is misleading at best. The authors need to specify for both fold changes (log2, x-axis) and significance (log10 p-value, y-axis). P < 0.05? P < 0.001? The authors highlighted significantly enriched proteins in the supplementary information, and P < 0.05, so I suggest including that in the manuscript. Also, running a statistical test on only 2 samples in a group is not advised. I strongly suggest increasing the sample size.
Reviewer 2 Report
Overall a novel approach to try and understand why CAG repeats expand, but there are a few technical shortcomings.
1. The use of Xenopus extracts: there is no data in CAG instability in xenopus species, it really should have been done with human cell extracts.
2. The CAG loop the engineered is only 7 CAG, which is far smaller than the typical human wild-type alleles and nowhere near the threshold of 37 repeats needs to see CAG expansion in humans, as less than 37 repeats are seen to be stable.
3. The context of CAG needs to be in place in the HTT exon, there is a massive assumption flanking sequences have no impact.
4. There is no validation of any of the proteins defined as enriched by biochemical methods of defining protein-DNA interactions.
This is too preliminary for publication. With human extracts, >37CAG constructs in exon1 DNA context and some EMSA validation, this could be a great manuscript.
Lastly, the discussion speculation about prions and one yeast model study using a massively overexpressed fragment of huntingtin makes no sense. Human PBMC and human GWAS data establish DNA repair defects, leading to CAG expansion, decades before we see disease onset or the presence of aggregated protein (which has never been seen in neurons primarily lost in HD) , and many HD phenotypes in HD-derived human cells take place without any aggregation. At this point aggregation is a phenomenon linked entirely to fragment molar massive overexpression of clinically irrelevant hyper-alleles of CAG.
Human GWAS definitively demonstrated that proteostatic pathways and mechanisms were not relevant to HD age of onset or severity, so to try and explain the data on an outdated theory that has always been a weak association to an amyloid-like hypothesis is not optimal.
It's well written.
Round 2
Reviewer 1 Report
The authors have addressed problems in the introduction, but why -log(p-value) of 0.4 in other words, the p-value of 1.3 was used for significance was not explained. Without a proper power calculation and justification, it is advised to use a p-value of 0.05 or a -log(p-value) of 1.3. Revise Figure 5 using 1.3 as the cut-off.
Reviewer 2 Report
I would like to see a small paragraph about the technical aspects of the assay that could be changed in future work: the use of human cell extracts and the context of CAG expansion in the disease protein exon, as well as the interactome changes with CAA alleles and the use of short and long expansions of CAG beyond the disease threshold of 37 CAG. It could be presented as a paragraph of "future applications".
Round 3
Reviewer 1 Report
The authors adequately addressed all the concerns.
I have two minor comments.
Please add a comma before "it" on line 58.
On line 58, some patients with 36-39 do develop HD, and patients with 40+ is considered as adult on-set while 60+ is considered adult-onset (Keum et al., 2016, The American J of Human Genetics).
Thank you for your scientific contribution.